# Long-range spatial extension of exciton states in van der Waals heterostructure

Zhiwen Zhou ⓘ, E. A. Szwed, W. J. Brunner, H. Henstridge, L. H. Fowler-Gerace & L. V. Butov ⓘ ✉

Narrow lines in photoluminescence (PL) spectra of excitons are characteristic of low-dimensional semiconductors. These lines correspond to the emission of exciton states in local minima of a potential energy landscape formed by fluctuations of the local exciton environment in the heterostructure. The spatial extension of such states was in the nanometer range. In this work, we present studies of narrow lines in PL spectra of spatially indirect excitons (IXs) in a MoSe$_2$/WSe$_2$ van der Waals heterostructure. The narrow lines vanish with increasing IX density. The disappearance of narrow lines correlates with the onset of IX transport, indicating that the narrow lines correspond to localized exciton states. The narrow lines extend over distances reaching several micrometers and over areas reaching ca. ten percent of the sample area. This macroscopic spatial extension of the exciton states, corresponding to the narrow lines, indicates a deviation of the exciton energy landscape from random potential and shows that the excitons are confined in moiré potential with a weak disorder.

Narrow lines with linewidths $\lesssim 1$ meV in PL spectra of excitons are ubiquitous in low-dimensional semiconductors. The narrow lines found in a GaAs quantum dot correspond to the emission of exciton states in the dot[1]. The narrow lines observed in GaAs quantum wells originate from the emission of exciton states in local minima of random potential formed by fluctuations of the local exciton environment in the heterostructure, e.g., quantum well width and materials fluctuations[2–5].

Recent studies revealed narrow lines in PL spectra of excitons in van der Waals (vdW) heterostructures composed of single atomic layers of transition-metal dichalcogenides (TMD). The narrow lines observed in a TMD electrostatically defined trap correspond to the emission of exciton states in the trap[6]. The narrow lines were also observed for excitons in local minima of random potential in monolayer TMD[7–11] and for excitons confined by strain in the regions of the heterostructure flake edges[12], heterostructure wrinkles[13], or nanopillars[14–16].

Local minima in the exciton potential landscape can be also formed in moiré superlattices in TMD heterostructures[17–21]. The moiré potentials can be affected by atomic reconstruction[22–24]. Narrow lines were observed for excitons in bilayer TMD heterostructures with moiré potentials[25–33].

Similar to other low-dimensional semiconductors, such as GaAs heterostructures outlined above, narrow lines are ubiquitous in vdW heterostructures. In addition to TMD heterostructures with the exciton confinement caused by electrostatic traps[6], random potentials[7–11], or strain[12–16] and TMD bilayer heterostructures with moiré potentials[25–33] outlined above, narrow lines were also observed in TMD bilayer heterostructures where moiré potentials are suppressed by hBN spacers[34] and in TMD trilayer heterostructures[35].

In contrast to random potentials, which are characteristic of semiconductor heterostructures with 2D layers formed by several monolayers (like GaAs heterostructures) or single monolayers (like TMD heterostructures), moiré potentials are periodic in the heterostructure plane. Their lateral period is typically in the range of ca. 10 nm, exceeding the exciton Bohr radius ca. 1 nm and providing a confining potential for excitons[17–21]. Narrow lines are observed both in heterostructures with moiré potentials and in heterostructures without moiré potentials, as outlined above, and both these types of heterostructure have disorder potentials. Since both disorder and moiré

Department of Physics, University of California San Diego, La Jolla, CA, USA. ✉e-mail: lvbutov@physics.ucsd.edu

potentials can produce the narrow-line emission, the roles of the disorder and moiré potentials in the origin of the narrow lines remain unclear, as outlined, in particular, in recent studies of TMD heterostructures[34].

The major difference between moiré potentials and disorder potentials in the origin of the narrow lines is a spatial ordering for the former. The lateral extension of the narrow lines is given by the lateral extension of the corresponding exciton states. The extension of localized exciton states in a random potential is typically on the order of nanometers, and even the extension of delocalized exciton states, given by the mean free path, is typically in the nanometer range for excitons in semiconductor heterostructures, as overviewed in ref. 36. In contrast, exciton states in a periodic lattice potential, such as a moiré potential, can extend over long distances limited by imperfections of the periodic potential in the heterostructure.

A strong disorder potential both limits the spatial extension of localized exciton states and reduces the diffusivity of delocalized excitons. Therefore, heterostructures with longer extension of localized exciton states may provide more efficient exciton transport with higher diffusivity and mean free path. The long lifetimes of IXs give an opportunity to realize long-range efficient exciton transport, and IX transport in TMD heterostructures is intensively studied[36-58]. Recent studies show the long-range IX transport[57], the long-range IX mediated spin transport[58], and the high IX diffusivity and mean free path[36]. In this work, we verify, in particular, if the heterostructure presenting this efficient exciton transport[36,57,58] is characterized by a long spatial extension of localized exciton states.

In earlier studies of narrow lines, outlined above, the spatial extension of the narrow lines was limited by the spatial resolution of the optical experiments, ca. 1 μm. This short extension of exciton states associated with the narrow lines is characteristic of disordered potentials as shown, in particular, in studies of narrow lines in GaAs heterostructures[2-5] where moiré potentials are absent and narrow lines originate from the emission of exciton states in local minima of random potential in the heterostructure. In this work, we studied narrow lines in PL spectra of IXs in a MoSe$_2$/WSe$_2$ vdW heterostructure. We observed that the narrow lines extend over distances reaching several microns. This macroscopic spatial extension of the exciton states, corresponding to the narrow lines, indicates a deviation of the excitonic energy landscape from random potential. An ordering in the local environment of excitons, such as a moiré potential disordered weakly, is consistent with the observed long-range spatial extension of the exciton states. The long-range extension of exciton states facilitates efficient exciton transport in the heterostructure.

## Results and discussion

In the studied MoSe$_2$/WSe$_2$ vdW heterostructure, the adjacent MoSe$_2$ monolayer and WSe$_2$ monolayer form the separated electron and hole layers and IXs are formed by electrons and holes confined in these separated layers[59]. Twisting between the MoSe$_2$ and WSe$_2$ monolayers with the twist angle $\delta\theta$ ~ 1.1° produces a moiré potential with the moiré superlattice period $b$ ~ $a/\delta\theta$ ~ 17 nm ($a$ is the lattice constant)[17-21]. The heterostructure details and the optical measurements are outlined in Supplementary Information (SI).

### Energies of narrow-line exciton states

Figure 1 shows narrow lines in the PL spectra of IXs in the heterostructure. With increasing density, the energies of the narrow lines stay fixed (Fig. 1), more data on the density dependence of the narrow lines is presented in Fig. S2 in the SI. This indicates that each narrow line corresponds to an exciton state with a low sensitivity to the average exciton density. An exciton in a moiré cell with certain occupations of neighbor cells is consistent with such a state[25-33]. For such exciton states, adding an exciton to a neighbor cell leads to an increase by the inter-cell interaction energy exceeding the linewidth of the narrow line

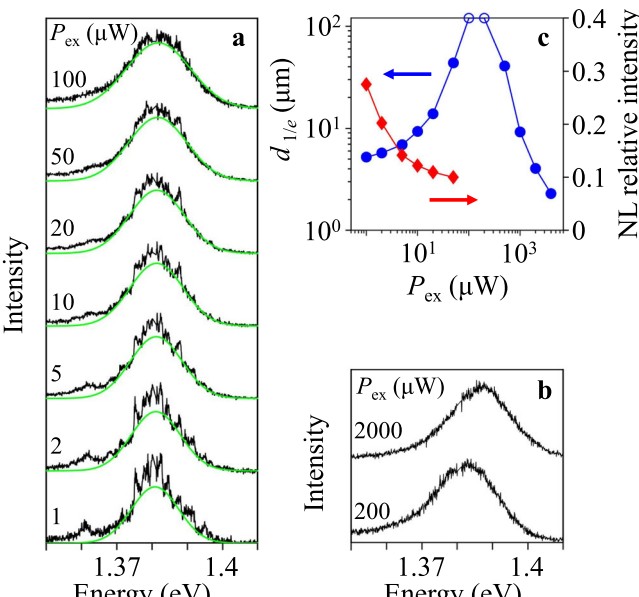

**Fig. 1 | Narrow lines in PL spectra of IXs. a**, **b** The excitation power $P_{ex}$ dependence of IX spectra. The narrow lines are observed on a background of a broad line approximated by a Gaussian (the green line) for each spectrum. The laser excitation spot is focused to a spot ~2 μm in diameter. **c** Comparison of the relative intensity of the narrow lines (NLs) in the PL spectra with IX transport in the heterostructure. The former is presented by the ratio of the sum of spectrally integrated intensities of the narrow lines to the spectrally integrated intensity of the broad line in the PL spectrum (red diamonds), and the latter is presented by the 1/$e$ decay distance of IX transport $d_{1/e}$ from ref. 57 (blue points). $T$ = 3.5 K. The disappearance of narrow lines in the spectrum correlates with the onset of IX transport.

so, at low densities, cells with statistically distinct occupations of the neighbor cells can produce narrow PL lines with the lack of continuous energy shift with density[25-33]. In contrast, statistical averaging over different exciton states gives the broad PL line with the energy monotonically increasing with average exciton density $n$ (Figs. 1 and S2 in SI). This average energy shift can be approximated by the mean field "capacitor" formula $\delta E = 4\pi e^2 d_z n/\varepsilon$[60] ($d_z$ ~ 0.6 nm is the separation between the electron and hole layers and $\varepsilon$ ~ 7.4 is the dielectric constant for the heterostructure[61]) and, in particular, can be used for estimating $n$.

### Disappearance of narrow lines and onset of exciton transport

The narrow lines vanish with increasing density and, at high densities, a broad PL line dominates the spectrum (Fig. 1a, b). In Fig. 1c, the relative intensity of the narrow lines in the PL spectra is compared with IX transport in the heterostructure. The former is presented by the ratio of the sum of spectrally integrated intensities of the narrow lines to the spectrally integrated intensity of the broad line in the PL spectrum, and the latter is presented by the 1/$e$ decay distance of IX transport $d_{1/e}$ measured in ref. 57. The opportunity to achieve with varying density both IX localization and long-range IX transport, studied in ref. 57, enables such comparison. This comparison shows that the disappearance of narrow lines with increasing density in the range of the pump power $P_{ex}$ ~ 1–50 μW correlates with the onset of IX transport (Fig. 1c). The anticorrelation with transport indicates that the narrow lines correspond to localized excitons. However, this anticorrelation does not establish the nature of localization that may be caused by a disorder potential or by an ordered moiré potential in the heterostructure.

Figure 1 shows that the narrow lines vanish with the onset of IX transport, however, they do not re-appear at the higher densities where IX re-entrant localization, outlined in ref. 57, is observed. This is

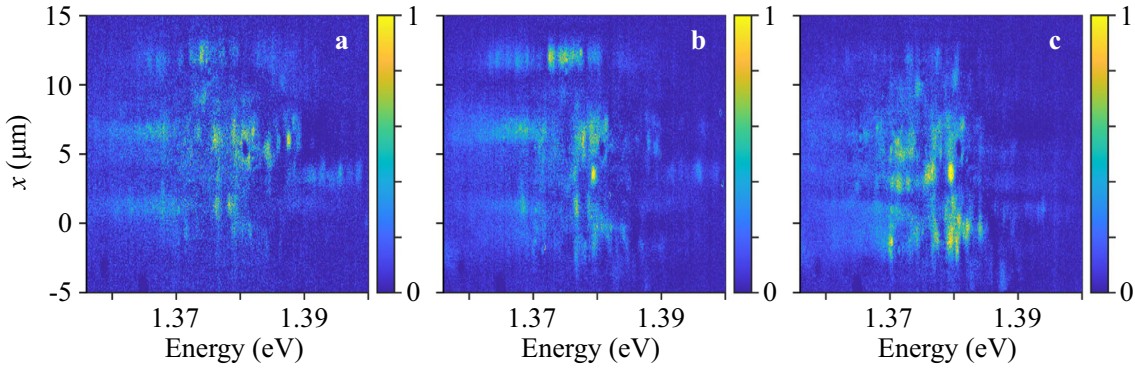

**Fig. 2 | x-energy maps of narrow lines.** x-energy maps of exciton PL for $y = -3.5$ μm (**a**), $y = -2.2$ μm (**b**), and $y = -0.9$ μm (**c**). x-energy maps for other $y$ locations in the heterostructure are shown in Fig. S7 in SI. The signal is integrated within 1.3 μm in the $y$ direction. The broad background (given by Gaussians in Fig. 1a) is subtracted. x-energy maps without background subtraction are shown in Fig. S8 in SI. The excitation spot is defocused over a spot ~25 μm in diameter covering the heterostructure area for a weak excitation of the entire sample. The excitation power of this defocused excitation is 50 μW. $T = 4.2$ K. Narrow lines with longer extensions along $x$ are seen as vertical modulated lines in the x-energy maps. The narrow lines and, in turn, the corresponding exciton states, seen in the x-energy maps, extend over long distances reaching several micrometers.

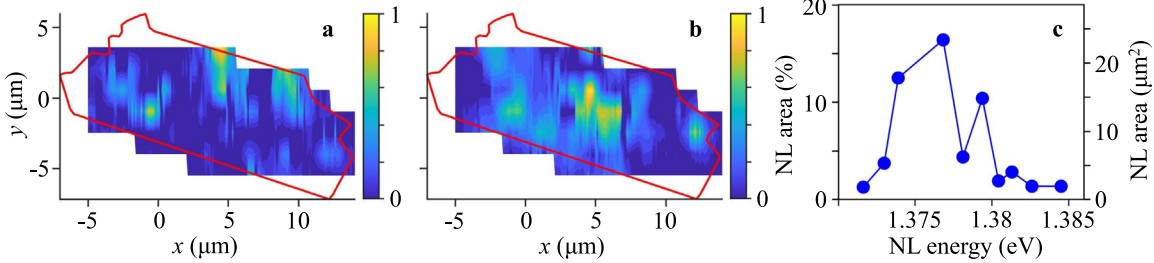

**Fig. 3 | x-y maps showing the spatial extension of exciton states corresponding to narrow lines.** x-y maps of spatial extension of the narrow line at $E = 1.3739$ eV (**a**) and the narrow line at $E = 1.3768$ eV (**b**). The signal is collected above the broad background (given by Gaussians in Fig. 1a) within the 1 meV linewidth of the narrow line. x-y maps for other narrow lines are shown in Fig. S10 in SI. The boundary of the MoSe$_2$/WSe$_2$ heterostructure is shown by the red line. The excitation spot is defocused over a spot ~25 μm in diameter covering the heterostructure area for a weak excitation of the entire sample. The excitation power of this defocused excitation is 50 μW. $T = 4.2$ K. **c** The area of the exciton state corresponding to the narrow line (NL) vs. the energy of the narrow line. The area boundaries are defined by 1/$e$ drop of the narrow-line intensity. The area percentage of the entire measured sample area is also shown. The narrow lines and, in turn, the corresponding exciton states seen in the x-y maps extend over macroscopic areas reaching ca. ten percent of the measured sample area.

consistent with the narrow line association with the exciton localization in local minima of a potential energy landscape formed by variations of the local exciton environment in the heterostructure. The re-entrant localization at the higher densities due to insulating phase, such as the Mott insulator and the Bose glass[62], is of a different origin. In particular, for the higher densities when most of the moiré cells are occupied, the particle transport from cell to cell leads to double occupancy that creates a gap for particle-hole excitations, consequently making the state insulating[62]. Figure 1 shows that narrow lines are not characteristic of this high-density insulating phase.

### g factor of narrow-line exciton states

For all narrow lines, the measured excitonic g factor is $g \approx -15.5 \pm 0.7$ as shown in Fig. S6 and outlined in SI. Excitonic g factor is determined by the local atomic registry and the measured g factor corresponds to $H_h^h$ site in the moiré potential of the MoSe$_2$/WSe$_2$ heterostructure with, in turn, $H$ stacking[17–20,25,63]. For TMD heterostructures with moiré potentials, a coincidence of g factor for all narrow lines was found in ref. 25. The g factor specific for a certain local atomic registry ($H_h^h$ in our case) shows that the narrow lines correspond to excitons in the specific site ($H_h^h$ in our case) of the moiré potential.

The same local atomic registry $H_h^h$ may extend over a considerable part of the sample in (reconstructed) moiré potentials that makes the measured g factor essentially insensitive to the location of the exciton state in the sample[63]. In particular, for the exciton Bohr radius much smaller than the moiré site, excitons can be localized by random

potential fluctuations within the moiré site as outlined in ref. 34. Therefore, the same and atomic-registry-specific g factor of narrow lines is insufficient to establish the nature of localization of the corresponding exciton states that may be caused by a strong disorder or by an ordered moiré potential in the heterostructure.

### Spatial extension of narrow-line exciton states

Figure 2 shows x-energy maps of the exciton PL. In these maps, the narrow lines are revealed by the spectrally narrow enhancements of the PL intensity. Figure 2 shows that narrow lines and, in turn, the corresponding exciton states, can extend over long distances reaching several micrometers.

Figure 2 shows the extension of the narrow-line exciton states in the x-direction. The measured x-energy maps at different $y$ locations allow building the x-y maps for the exciton states corresponding to the narrow lines. Examples of the x-y maps for the exciton states are presented in Fig. 3. (Figures S7 and S10 in SI show x-energy maps for all measured $y$, covering essentially the entire heterostructure area, and x-y maps for many of the narrow-line exciton states seen in the x-energy maps). Figure 3 shows that the narrow lines extend over distances reaching several micrometers and over areas reaching ca. ten percent of the measured sample area.

The observed macroscopic spatial extension of exciton states, corresponding to the narrow lines, indicates a deviation of the exciton energy landscape from random potential. A strong disorder potential does not produce macroscopically extended localized exciton states.

In particular, no such extension was observed in any semiconductor system, including GaAs and vdW heterostructures outlined in the introduction, where narrow lines originate from the emission of exciton states localized in local minima of random potential formed by fluctuations of the local exciton environment in the heterostructure, e.g., stress, dielectric, electrostatic, and materials fluctuations.

In turn, the observed macroscopic spatial extension of localized exciton states, corresponding to the narrow lines, indicates ordering in the local environment of excitons: The exciton state at a certain energy, corresponding to the narrow line, extends over macroscopic length and area (Figs. 2 and 3). A moiré potential is consistent with such ordering and long-range spatial extension of localized exciton states. We note that different narrow lines and their corresponding exciton states are extended over different regions of the heterostructure (e.g., compare the regions for different narrow lines in Fig. 3a, b). This indicates that the local environment for excitons fluctuates over the heterostructure, however, the fluctuations are small enough to allow the long-range extension of the individual exciton states. Therefore, the long-range extension of the narrow lines shows that the excitons are confined in moiré potential with a weak disorder.

Moiré potentials with a weak disorder can host long-range ballistic exciton transport due to exciton superfluidity in periodic potentials[62]. A strong disorder destroys superfluidity[62]. Therefore, the weakness of disorder in the moiré potential, revealed by the long-range extension of the exciton states (Figs. 2 and 3), suggests an opportunity to observe the long-range ballistic transport of excitons in this weakly disordered moiré potential. The studies of exciton transport in the same heterostructure show the long-range IX transport[57], the long-range IX mediated spin transport[58], and the high IX diffusivity and mean free path[36].

The extension of exciton states over distances reaching several micrometers raises a question of distinguishing such states from delocalized excitons states and a question if such extended states can be called localized states. In this work, we qualitatively discuss an exciton state confined in a region, even of a large area, as a localized state and exciton states, which can travel over different localization regions, as delocalized states. The long-range extension of localized exciton states and their small energy difference facilitates exciton transport over different localization regions in the heterostructure.

In summary, we studied narrow lines in PL spectra of IXs in a MoSe$_2$/WSe$_2$ heterostructure. We found that the disappearance of narrow lines correlates with the onset of IX transport, indicating that the narrow lines correspond to localized exciton states. We found that the narrow lines extend over distances reaching several micrometers and over areas reaching ca. ten percent of the sample area. This macroscopic spatial extension of exciton states, corresponding to the narrow lines, indicates a deviation of the exciton energy landscape from random potential and shows that the excitons are confined in moiré potential with a weak disorder. The long-range extension of exciton states facilitates efficient exciton transport.

## Methods

### Van der Waals heterostructure
The MoSe$_2$/WSe$_2$ heterostructure was assembled using the dry-transfer peel technique[64]. The heterostructure details are presented in the SI.

### Optical measurements
Excitons were generated by a continuous-wave Ti:sapphire laser with the excitation energy $E_{ex}$ = 1.689 eV resonant to DX in the WSe$_2$ heterostructure layer (a cw semiconductor laser with similar $E_{ex}$ was used for the data in Figs. S3b–d, S4, S5, and S12 in SI). PL spectra were measured using a spectrometer with a resolution of 0.2 meV and a liquid-nitrogen-cooled CCD. A spectrometer slit along the $x$ direction was used to select a $\delta y$ = 1.3 μm wide portion of the sample emission.

An emission that is outside the selected $\delta y$ = 1.3 μm area did not enter the spectrometer and, therefore, was separated. The spectrometer showed on the CCD detector the $x$-energy map, where the $x$-axis is along the spectrometer slit and the orthogonal axis on the CCD detector shows the spectrum for each $x$ position. For the $x$-energy maps in Figs. 2, S3, S4, and S7–S9 in SI, we used a defocused laser excitation covering the entire heterostructure that allowed obtaining the $x$-energy maps without moving and precise positioning of the laser beam. The $x$-energy images were measured with the step 1.3 μm and the signal integration within 1.3 μm in the $y$ direction given by the slit (the slit positions in the measurements of the $x$-energy maps are shown in Fig. S7h in SI). For the $x$-$y$ maps, the step in the slit position and the signal integration within the slit 1.3 μm gave the spatial resolution in the $y$ direction. NA = 0.64 of the lens gave the spatial resolution 0.7 μm in the $x$ direction.

## Data availability
The data are available via Figshare at https://doi.org/10.6084/m9.figshare.30944516.

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

## Acknowledgements

We thank M.M. Fogler for discussions and A.K. Geim for teaching us manufacturing TMD heterostructures. The experiments were supported by the Department of Energy, Office of Basic Energy Sciences, under award DE-FG02-07ER46449 (L.V.B.). The heterostructure manufacturing and data analysis were supported by NSF grants 1905478 (L.V.B.) and 2516006 (L.V.B.).

## Author contributions

L.V.B. designed the project. L.H.F.-G. manufactured the TMD heterostructure. Z.Z., E.A.S., W.J.B., and H.H. performed the measurements. Z.Z. and L.V.B. analyzed the data. L.V.B. wrote the manuscript with inputs from all the authors.

## Competing interests

The authors declare no competing interests.

## Additional information

**Supplementary information** The online version contains Supplementary material available at https://doi.org/10.1038/s41467-026-70218-4.

