## [Transparent Peer Review File · Nature Communications]

Long-range spatial extension of exciton states in van der Waals heterostructure

Corresponding Author: Professor Leonid Butov

Version 0:

Reviewer comments:

Reviewer #1

(Remarks to the Author)

The manuscript by Zhou et al. entitled “Long-range spatial extension of exciton states in van der Waals heterostructure” studies narrow lines in the PL emission of Interlayer Excitons in a hBN encapsulated MoSe₂/WSe₂ heterostructure. The work demonstrates narrow line PL emission extending over several microns. The authors attribute the long-range narrow line formation to an ordered varying potential landscape allowing for consistent trapping of IXs at potential minima. The data is good quality, the paper is well written, and the long-range mapping of narrow line energies is intriguing. However, given the large body of work that already exist on moiré trapped excitons in twisted TMD heterostructures, I have some concerns about the papers methodology and impact that should be addressed before recommendation for publication in Nature Communications.

I am unclear about the authors methodology in this experiment. The authors state “The x–Energy images were measured with the step 1.3 μm and the signal integration within 1.3 μm in the y direction given by a slit (the slit positions in the measurements of the x–Energy maps are shown in Fig. S4h).” I interpret this to mean that the authors used a slit to image only a 1.3 μm wide portion of their sample emission along y. They then must have stepped their laser in 1.3 μm steps in x and measured the PL emission on their CCD. If this is correct, then it is unclear to me how the authors have adequately separated any extraneous emission that is outside the 1.3 μm area they claim to measure in x. If this is incorrect, then I believe the authors should revise their language and include a figure demonstrating their optical set up.

One concern I have is in the way the authors characterize the disappearance of their narrow lines with power. Figure 1 c only shows the integrated narrow line emission divided by the integrated broader gaussian peak; however, it would make sense that the narrow lines scale slower with power than the broader emission. The authors should also show the total integrated emission from the narrow lines and gaussian subtracted, position dependent narrow line emission for higher powers like in Figure 2.

I have a concern about the characterization of the narrow lines as moiré trapped exciton emission. The authors vary their excitation power to control their exciton density and point to the disappearance of the narrow line emission at high power along with the onset of IX transport in reference [40] as evidence of the moiré becoming over filled. However, reference [34] demonstrates narrow line emission in a heterostructure where the moiré is suppressed. Because of this, the validity of the main conclusion of this paper is dependent on the interpretation that large spatial extent narrow line emission can only be caused by a moiré. This argument could be strengthened by the introduction of a control device. If the authors are correct in their interpretation, the narrow lines present in an hBN separated heterostructure would not remain over a few microns and it would be easier to attribute these results to a moiré rather than another ordered or coherent state.

I also have a concern about the reproducibility of these results. The few micron extent of the narrow line emission is a unique and interesting result. However, given the phenological nature of the results presented in this paper, similar results should be shown over multiple samples, and a lack of such results should be shown on a control as discussed in the previous comment to sufficiently convince the reader of this papers conclusion.

A minor comment I have pertains to figure and reference presentation. The figures have inconsistent letter placement, for example in Figure 1, the letters a and b appear in the top right of the plots and c appears in the top left. In reference [42] superfluid is misspelled. The authors should double check for errors.

Reviewer #2

(Remarks to the Author)

Reviewer #3

(Remarks to the Author)

In this manuscript, the authors probed the spatial extension of narrow PL emission lines of interlayer excitons (IX) in MoSe₂-WSe₂ heterobilayer at cryogenic temperatures. By showing the correlation between the disappearance of narrow PL lines and the onset of IX transport, they showed that narrow lines originate from localized exciton states. Additionally, the observed spatial extension of the exciton states over distances of several micrometers was attributed to the localization of excitons in moiré potentials with weak disorder, rather than the random potential.

While the demonstration of the spatial extension of narrow emission lines over several micrometers is interesting and important for obtaining periodic quantum emitters from these heterobilayers, the results/evidences and discussion provided by the authors to support their claim are not sufficient enough for the publication of the manuscript in its current version in Nature Communications.

In the following, I give some additional comments and questions:

1. Temperature-dependent PL measurements need to be performed along the extended region of the narrow emission lines to see whether they show the same behavior or not at different locations. (Major)
2. How do the lifetimes of localized interlayer excitons change spatially in the extended region of the narrow lines? In the case of moiré potentials with weak disorder, one can expect to obtain a uniform lifetime along the extended region. Did the authors measure the lifetimes along the extended region of the narrow lines? The position-dependent lifetime mapping would also be valuable in further strengthening their claim of having moiré potentials with weak disorder. (Major)
3. The presence of uniformity of the moiré pattern should be verified through one of the imaging techniques, such as TEM, piezoresponse force microscopy, etc. (Major)

Reviewer #4

(Remarks to the Author)

This paper appears to be a sequel to a series of publications by the same group of authors [1. Nature Communications (2024) 15:9454; 2. Nature Photonics (2024) 18, 823]. The subject matter studied is of high interest to the scientific community. I would support the publication of this manuscript if the authors can provide additional information to clarify the following issues:

1. In the introduction, the authors should cite their previously published papers on the same subject and explain what new information the current study provides. This will give readers a clearer perspective on the continued development of this research direction.
2. The authors state that "the disappearance of narrow lines correlates with the onset of IX transport." Please specify the pump power at which this correlation occurs.
3. The caption of Fig. 1 mentions that "a higher-energy IX line appearing at ~1.41 eV at high IX densities was attributed to the appearance of moiré cells with double occupancy in Ref. [41]." Please explicitly mark the ~1.41 eV line in Fig. 1, as it is not easily identifiable to readers.
4. If the narrow lines observed several microns away from the excitation spot correspond to the same exciton states, then their energies should be nearly identical, and the PL line shapes at different positions should correlate with the center-of-mass wavefunction of the excitons in the periodic moiré potential. The authors should provide a figure (as supplementary information) showing detailed line shapes of all narrow lines within the relevant spectral range at various x positions. This will allow readers to examine whether the lines indeed originate from the same exciton states. Such analysis will also be valuable for future theoretical work aiming to model PL line shapes and exciton diffusion characteristics in this system.
5. To better understand the influence of exciton diffusion from a tightly focused excitation spot, the authors should compare PL maps obtained using a 2- μ m illumination spot with PL maps obtained under uniform illumination (i.e., an excitation spot significantly larger than 10 μ m).

Version 1:

Reviewer comments:

Reviewer #1

(Remarks to the Author)

The authors responded to all of our comments. The paper is now suitable for publication.

Reviewer #2

(Remarks to the Author)

Reviewer #3

(Remarks to the Author)

COMMENTS TO THE AUTHORS

I thank the authors for their efforts in addressing my questions and comments. They have performed temperature dependent-PL and lifetime measurements to address my first and second questions. I have one remaining minor comment related to my first comment. However, my third comment, as detailed below has not yet been adequately addressed and still needs to be resolved before I recommend the publication of current manuscript in Nature Communications.

My additional comments and questions:

1. For my first comment, authors performed the temperature-dependent PL measurements in the temperature range of 1.7 K to 10 K. They should also discuss their results in terms of the linewidths of the narrow emitter lines at each position to see whether they are correlated. (Minor)
2. Complementary to the fourth comment of Reviewer 1, to answer also my third comment, authors need to fabricate another sample with (thin ~2nm) or without top hBN layer, and perform one of the characterization techniques I suggested to show the uniformity of the moiré pattern. In that way, they can also show the reproducibility of their results in another sample. It might also help to measure the lifetime of individual narrow PL lines at different positions as suggested before.

Reviewer #4

(Remarks to the Author)

The authors have answered all my questions. I am satisfied with the revised manuscript and supplemental material. I would support the publication of the revised version in Nature Communications.

RESPONSE TO REVIEWERS' COMMENTS

Response to Remarks of Reviewer 1 (Reviewer's comments are marked by italic font, changes in the manuscript are marked by blue font, references correspond to the revised manuscript).

The manuscript by Zhou et al. entitled "Long-range spatial extension of exciton states in van der Waals heterostructure" studies narrow lines in the PL emission of Interlayer Excitons in a hBN encapsulated MoSe₂/WSe₂ heterostructure. The work demonstrates narrow line PL emission extending over several microns. The authors attribute the long-range narrow line formation to an ordered varying potential landscape allowing for consistent trapping of IXs at potential minima. The data is good quality, the paper is well written, and the long-range mapping of narrow line energies is intriguing. However, given the large body of work that already exist on moiré trapped excitons in twisted TMD heterostructures, I have some concerns about the papers methodology and impact that should be addressed before recommendation for publication in Nature Communications.

We thank Reviewer 1 for a high estimate of our work and useful remarks. We address remarks of Reviewer 1 in the revised manuscript as outlined below.

I am unclear about the authors methodology in this experiment. The authors state "The x-Energy images were measured with the step 1.3 μm and the signal integration within 1.3 μm in the y direction given by a slit (the slit positions in the measurements of the x-Energy maps are shown in Fig. S4h)." I interpret this to mean that the authors used a slit to image only a 1.3 μm wide portion of their sample emission along y. They then must have stepped their laser in 1.3 μm steps in x and measured the PL emission on their CCD. If this is correct, then it is unclear to me how the authors have adequately separated any extraneous emission that is outside the 1.3 μm area they claim to measure in x. If this is incorrect, then I believe the authors should revise their language and include a figure demonstrating their optical set up.

Yes, this is correct (with a clarification that for x-Energy maps in Fig. S4 that is Fig. S7 in the revised manuscript, we used defocused laser excitation covering the entire sample so stepping the laser beam was not needed). The 1.3 μm wide portion of the sample emission in the y direction was selected by the spectrometer slit and *any extraneous emission that is outside the 1.3 μm area* did not enter the spectrometer and, therefore, was separated. As to the x direction, the emission from different x positions on the sample appear at different x locations on the CCD detector. To address this remark of Reviewer 1, we include an additional clarification of the optical measurements in SI: "A spectrometer slit along the x direction was used to select a $\delta y = 1.3 \mu\text{m}$ wide portion of the sample emission. An emission that is outside the selected $\delta y = 1.3 \mu\text{m}$ area did not enter the spectrometer and, therefore, was separated. The spectrometer showed on the CCD detector the x-Energy map where the x axis is along the spectrometer slit and the orthogonal axis on the CCD detector shows the spectrum for each x position. For the x-Energy maps in Fig. 2 in the main text and in Figs. S3, S4, S7-S9, we used a defocused laser excitation covering the entire heterostructure that allowed obtaining the x-Energy maps without moving and precise positioning of the laser beam."

One concern I have is in the way the authors characterize the disappearance of their narrow lines with power. Figure 1 c only shows the integrated narrow line emission divided by the integrated broader gaussian peak; however, it would make sense that the narrow lines scale slower with power than the broader emission. The authors should also show the total integrated emission from the narrow lines and gaussian subtracted, position dependent narrow line emission for higher powers like in Figure 2.

Indeed, *the narrow lines scale slower with power than the broader emission*. To address this remark of Reviewer 1, we add in SI new Fig. S3 showing the total integrated emission from the narrow lines (in Fig. S3a) and gaussian subtracted, position dependent narrow line emission for higher powers like in Figure 2 (in Figs. S3b-d, we performed these measurement to address this remark of Reviewer 1). We

also add the corresponding description of this new figure: “Figure 1c in the main text shows the relative intensity of the narrow lines presented by the ratio of the sum of spectrally integrated intensities of the narrow lines to the spectrally integrated intensity of the broad line in the PL spectrum. Figure S3a shows the sum of spectrally integrated intensities of the narrow lines and shows the spectrally integrated intensity of the broad line in the PL spectrum. The intensities of narrow lines grow slower with power than the intensity of the broad line (Fig. S3a), consistent with the data in Fig. 1c in the main text. Figures S3b-d show x -Energy maps for higher excitation powers. With increasing density, the energies of the narrow lines stay fixed (Fig. 3b-d), consistent with the lack of change of energies of narrow lines with density discussed in the main text.”

I have a concern about the characterization of the narrow lines as moiré trapped exciton emission. The authors vary their excitation power to control their exciton density and point to the disappearance of the narrow line emission at high power along with the onset of IX transport in reference [40] as evidence of the moiré becoming over filled. However, reference [34] demonstrates narrow line emission in a heterostructure where the moiré is suppressed. Because of this, the validity of the main conclusion of this paper is dependent on the interpretation that large spatial extent narrow line emission can only be caused by a moiré. This argument could be strengthened by the introduction of a control device. If the authors are correct in their interpretation, the narrow lines present in an hBN separated heterostructure would not remain over a few microns and it would be easier to attribute these results to a moiré rather than another ordered or coherent state.

Our interpretation is that *large spatial extent* of narrow line emission is caused by a relatively small disorder. In contrast, heterostructures with a large disorder show a much shorter spatial extent of narrow line emission as outlined in the manuscript. In contrast to Ref. [34] where MoSe₂ and WeSe₂ monolayers are separated by hBN that suppresses the moiré potential, in our heterostructure, MoSe₂ and WeSe₂ monolayers are adjacent and twisting between the MoSe₂ and WSe₂ monolayers with the twist angle $\delta\theta \sim 1.1^\circ$ produces a moiré potential with the moiré superlattice period $b \sim a/\delta\theta \sim 17$ nm. This is confirmed by the measured g factor $g \sim 15.5$, which corresponds to H_h^h site of the moiré potential as outlined in our manuscript: The g factor specific for a certain local atomic registry (H_h^h in our case) shows that the narrow lines correspond to excitons in the specific site (H_h^h in our case) of the moiré potential. Narrow lines can appear for excitons localized in local minima of moiré potential or for excitons localized by random potential fluctuations within the moiré site as outlined in Ref. [34]. However, for a random potential the spatial extent of narrow line emission is short. The large spatial extent of narrow line emission indicates a deviation of the exciton energy landscape from random potential and shows that the excitons are confined in moiré potential with a weak disorder, as outlined in our manuscript.

GaAs heterostructures, where the moire potential is clearly absent, can serve as a *control device*, and, indeed, in GaAs heterostructures *the narrow lines do not remain over a few microns* that agrees with the remark of Reviewer 1. In response to this remark, we add a clarification in the manuscript: “In earlier studies of narrow lines, outlined above, the spatial extension of the narrow lines was limited by the spatial resolution of the optical experiments, ca. 1 μ m. **This short extension of exciton states associated with the narrow lines is characteristic of disordered potentials as shown, in particular, in studies of narrow lines in GaAs heterostructures [2-5] where moiré potentials are absent and narrow lines originate from the emission of exciton states in local minima of random potential in the heterostructure.**”

I also have a concern about the reproducibility of these results. The few micron extent of the narrow line emission is a unique and interesting result. However, given the phenological nature of the results presented in this paper, similar results should be shown over multiple samples, and a lack of such results

should be shown on a control as discussed in the previous comment to sufficiently convince the reader of this paper's conclusion.

We checked that the long-range extension of the narrow PL lines is reproducible in multiple repeated measurements. We agree that it would be good to show similar results *over multiple samples*. However, so far the long-range extension of the narrow PL lines was realized in one sample in this work. This work demonstrates the existence of the long-range extension of exciton states in TMD heterostructures. Studying this phenomenon in other samples is the subject for future works. Furthermore, in response to a remark of Reviewer 4, we added in the introduction citations of our previous works [1. *Nature Communications* (2024) 15:9454; 2. *Nature Photonics* (2024) 18, 823] and outlined the relation between long-range extension of localized exciton states and efficient exciton transport. The efficient exciton transport was also realized in one sample, the same sample where the long-range extension of the narrow PL lines was observed. In response to this remark of Reviewer 1, we add a clarification in SI: “So far, the long-range extension of the narrow PL lines and the corresponding exciton states was realized in one sample in this work. Other studies of GaAs and TMD heterostructures show short extension of the narrow lines, below the spatial resolution of the optical measurements, as outlined in the main text. A shorter extension of exciton states likely originates from stronger disorder. This work demonstrates the existence of the long-range extension of exciton states in TMD heterostructures. Studying this phenomenon in other samples is the subject for future works. As outlined in the introduction in the main text, the long-range extension of the narrow PL lines is observed in the heterostructure presenting the long-range IX transport [2], the long-range IX mediated spin transport [3], and the high IX diffusivity and mean free path [4]. So far, this efficient exciton transport [2-4] was realized in one sample. Studying this efficient exciton transport in other samples and, in particular, verifying its relation to the long-range extension of the narrow PL lines is also the subject for future works.”

A minor comment I have pertains to figure and reference presentation. The figures have inconsistent letter placement, for example in Figure 1, the letters a and b appear in the top right of the plots and c appears in the top left. In reference [42] superfluid is misspelled. The authors should double check for errors.

We thank Reviewer 1 for noting this. We move all letters to the top right of the plots, double check for errors, and correct typos in the manuscript.

Response to Remarks of Reviewer 2.

We thank Reviewer 2 for the participation in reviewing our manuscript.

Response to Remarks of Reviewer 3 (Reviewer's comments are marked by italic font, changes in the manuscript are marked by blue font, references correspond to the revised manuscript).

In this manuscript, the authors probed the spatial extension of narrow PL emission lines of interlayer excitons (IX) in MoSe₂-WSe₂ heterobilayer at cryogenic temperatures. By showing the correlation between the disappearance of narrow PL lines and the onset of IX transport, they showed that narrow lines originate from localized exciton states. Additionally, the observed spatial extension of the exciton states over distances of several micrometers was attributed to the localization of excitons in moiré potentials with weak disorder, rather than the random potential.

While the demonstration of the spatial extension of narrow emission lines over several micrometers is interesting and important for obtaining periodic quantum emitters from these heterobilayers, the results/evidences and discussion provided by the authors to support their claim are not sufficient enough for the publicatin of the manuscript in its current version in Nature Communications.

We thank Reviewer 3 for noting the importance of *the demonstration of the spatial extension of narrow emission lines over several micrometers* presented in our work and for useful remarks. We address remarks of Reviewer 3 in the revised manuscript as outlined below.

In the following, I give some additional comments and questions:

1. Temperature-dependent PL measurements need to be performed along the extended region of the narrow emission lines to see whether they show the same behavior or not at different locations. (Major)

To address this remark of Reviewer 3, we performed *Temperature-dependent PL measurements*. These measurements show that the narrow emission lines show the same behavior at different locations. To outline this, we add in SI new Fig. S4 and its description: “x-Energy maps measured at different temperatures are shown in Fig. S4. These measurements show that the long-range extension of the narrow lines is also observed at higher temperatures, the energies of the narrow lines practically do not change with temperature, and the intensities of the narrow lines reduce with increasing temperature, similarly at different locations.”

2. How do the lifetimes of localized interlayer excitons change spatially in the extended region of the narrow lines? In the case of moiré potentials with weak disorder, one can expect to obtain a uniform lifetime along the extended region. Did the authors measure the lifetimes along the extended region of the narrow lines? The position-dependent lifetime mapping would also be valuable in further strengthening their claim of having moiré potentials with weak disorder. (Major)

To address this remark of Reviewer 3, we performed lifetime measurements with the available instrumentation. Unfortunately, in these measurements, the signal is insufficient to measure the position-dependent lifetime for a single selected narrow line. Therefore, we measured the position-dependent lifetime for spectrally integrated PL. The spectral integration combines in the measurements all narrow lines in the range $E < 1.46$ eV of the PL spectrum. Lifetimes of different narrow lines may be different. The *x*-Energy maps in the manuscript show the emergence of different narrow lines at different positions that may contribute to variations of the spectrally integrated lifetime along *x*. Our measurements show that these variations are rather small, roughly within the accuracy of the measurements. To outline the lifetime measurements, we add in SI new Fig. S5 and its description: “We measured the PL lifetime using excitation by a pulsed semiconductor laser and detecting the signal by APD. In these measurements, the signal is insufficient to measure the position-dependent lifetime for a single selected narrow line. Therefore, we measured the position-dependent lifetime for spectrally integrated PL (Fig. S5). The spectral integration combines in the measurements all narrow lines in the range \$E < 1.46\$ eV of the PL

spectrum. Lifetimes of different narrow lines may be different. The x -Energy maps in this work show the emergence of different narrow lines at different positions that may contribute to variations of the spectrally integrated lifetime along x . The measured lifetime varies roughly within the accuracy of the measurements along the heterostructure (Fig. S5). The measurements of the position-dependent lifetime for a single selected narrow line is the subject for future work.”

3. The presence of uniformity of the moiré pattern should be verified through one of the imaging techniques, such as TEM, piezoresponse force microscopy, etc. (Major)

We do not perform *TEM, piezoresponse force microscopy, etc.* measurements, in particular, because the studied sample is a unique sample, which shows the long-range IX transport [57], the long-range IX mediated spin transport [58], the high IX diffusivity and mean free path [36], as we outline in the revised introduction in response to a remark of Reviewer 4, and the long-range spatial extension of the narrow lines considered in this manuscript. We hesitate risking this unique sample by performing the above measurements, which may compromise the sample quality. Therefore, we limit the measurements by the optical measurements outlined in this manuscript. To address this remark of Reviewer 3, we add a clarification in SI: “This work demonstrates the existence of the long-range extension of exciton states in TMD heterostructures. Studying this phenomenon in other samples **and by other experimental techniques** is the subject for future work.”

Response to Remarks of Reviewer 4 (Reviewer's comments are marked by italic font, changes in the manuscript are marked by blue font, references correspond to the revised manuscript).

This paper appears to be a sequel to a series of publications by the same group of authors [1. Nature Communications (2024) 15:9454; 2. Nature Photonics (2024) 18, 823]. The subject matter studied is of high interest to the scientific community. I would support the publication of this manuscript if the authors can provide additional information to clarify the following issues:

We thank Reviewer 4 for a high estimate of our work and useful remarks. We address remarks of Reviewer 4 in the revised manuscript as outlined below.

1. In the introduction, the authors should cite their previously published papers on the same subject and explain what new information the current study provides. This will give readers a clearer perspective on the continued development of this research direction.

To address this remark of Reviewer 4, in the introduction, we add citations of our previous works [1. Nature Communications (2024) 15:9454; 2. Nature Photonics (2024) 18, 823] and outline the relation between long-range extension of localized exciton states and efficient exciton transport that *explain what new information the current study provides*: “A strong disorder potential both limits the spatial extension of localized exciton states and reduces the diffusivity of delocalized excitons. Therefore, heterostructures with longer extension of localized exciton states may provide more efficient exciton transport with higher diffusivity and mean free path. The long lifetimes of IXs give an opportunity to realize long-range efficient exciton transport and IX transport in TMD heterostructures is intensively studied [36-58]. Recent studies show the long-range IX transport [57], the long-range IX mediated spin transport [58], and the high IX diffusivity and mean free path [36]. In this work we verify, in particular, if the heterostructure presenting this efficient exciton transport [36,57,58] is characterized by a long spatial extension of localized exciton states.” “The long-range extension of exciton states facilitates efficient exciton transport in the heterostructure.” Refs. [57,58] in this addition are [1. Nature Communications (2024) 15:9454; 2. Nature Photonics (2024) 18, 823] noted by Reviewer 4. We also add citations of these works [57,58] and related clarification on p. 4: “the weakness of disorder in the moiré potential, revealed by the long-range extension of the exciton states (Figs. 2 and 3), suggests an opportunity to observe the long-range ballistic transport of excitons in this weakly disordered moiré potential. The studies of exciton transport in the same heterostructure show the long-range IX transport [57], the long-range IX mediated spin transport [58], and the high IX diffusivity and mean free path [36].”

2. The authors state that "the disappearance of narrow lines correlates with the onset of IX transport." Please specify the pump power at which this correlation occurs.

We add a specification of the pump power at which this correlation occurs: “the disappearance of narrow lines with increasing density in the range of the pump power $P_{\text{ex}} \sim 1 - 50 \mu\text{W}$ correlates with the onset of IX transport (Fig. 1c)”.

3. The caption of Fig. 1 mentions that “a higher-energy IX line appearing at ~ 1.41 eV at high IX densities was attributed to the appearance of moiré cells with double occupancy in Ref. [41].” Please explicitly mark the ~ 1.41 eV line in Fig. 1, as it is not easily identifiable to readers.

As shown in Ref. [58] (former Ref. [41]), the high-energy line is a broad line centered at ~ 1.41 eV and observed as a shoulder for the high-density spectra considered in this manuscript. Since the high-energy

line is not well seen in Fig. 1 and is not discussed in the text, we remove this sentence from the Fig. 1 caption. We thank Reviewer 4 for noting this sentence, which is not needed for this manuscript.

4. If the narrow lines observed several microns away from the excitation spot correspond to the same exciton states, then their energies should be nearly identical, and the PL line shapes at different positions should correlate with the center-of-mass wavefunction of the excitons in the periodic moiré potential. The authors should provide a figure (as supplementary information) showing detailed line shapes of all narrow lines within the relevant spectral range at various x positions. This will allow readers to examine whether the lines indeed originate from the same exciton states. Such analysis will also be valuable for future theoretical work aiming to model PL line shapes and exciton diffusion characteristics in this system.

To address this remark, we add in SI new Fig. S9 showing line shapes of narrow lines within the relevant spectral range at various x positions. We also add a description outlining the nearly identical narrow line energies at various x positions and outlining a possible use of these data for future theoretical work as noted by Reviewer 4: “Figures S9a and S9b show the x -Energy map and the corresponding spectra of narrow lines at various x positions. The lineshapes of the narrow lines are fit by the narrow Gaussians in Fig. S9c. Figure S9d shows the detailed spectra at various x positions for the narrow line at 1.3768 eV, which extends over long distances. The detailed lineshapes show a nearly identical energy of the narrow line at various x positions. The measured lineshapes may be useful for future theoretical work aiming to model PL line shapes, spectral diffusion, and exciton diffusion characteristics in this system.”

5. To better understand the influence of exciton diffusion from a tightly focused excitation spot, the authors should compare PL maps obtained using a 2- μm illumination spot with PL maps obtained under uniform illumination (i.e., an excitation spot significantly larger than 10 μm).

To address this remark of Reviewer, we add in SI new Fig. S12 comparing the x -Energy PL maps obtained using a *tightly focused excitation spot* and using defocused illumination of the entire sample. Similar to the measurements with the defocused laser excitation, the spectra obtained with the focused laser excitation show narrow lines and allow reconstructing x -Energy map of narrow lines, however the accuracy is lower. We outline this in the added description: “For the x -Energy maps in Fig. 2 in the main text and in Figs. S3, S4, S7-S9, we used a defocused laser excitation covering the entire heterostructure. The defocused laser excitation allows obtaining x -Energy maps of narrow lines without moving the laser excitation spot (Fig. S12a). For the data in Fig. S12b, we used a laser beam focused to a spot $\sim 2 \mu\text{m}$ in diameter and moved it in the x direction to obtain spectra for various x positions. Similar to the spectra obtained with the defocused laser excitation, the spectra obtained with the focused laser excitation show narrow lines, which are revealed by the spectrally narrow enhancements of the PL intensity (Fig. S12). The spectra obtained with the focused laser excitation at various x positions allow reconstructing an x -Energy map (Fig. S12b), however, the accuracy of this map is lower than the accuracy of the x -Energy map measured with the defocused laser excitation where moving and precise positioning of the laser beam is not needed (Fig. S12a).”

x -E energy PL maps obtained with the defocused excitation are more accurate and their density dependence (shown in new Fig. S3b-d) may also give relevant information regarding the *influence of exciton diffusion*. As shown in Fig. 1c and supported by Fig. S3, narrow lines vanish with increasing exciton diffusion and this is probably the strongest effect of exciton diffusion on narrow lines. Understanding the effect of exciton diffusion on narrow lines is the subject for future theoretical work.

A point-by-point response to the Reviewers' comments (Reviewer's comments are marked by italic font, changes in the manuscript are marked by blue font).

Reviewer #1 (Remarks to the Author):

The authors responded to all of our comments. The paper is now suitable for publication.

We thank Reviewer 1 for reviewing our manuscript and for useful comments.

Reviewer #2 (Remarks to the Author):

We thank Reviewer 2 for the participation in reviewing our manuscript.

Reviewer #3 (Remarks to the Author):

I thank the authors for their efforts in addressing my questions and comments. They have performed temperature dependent-PL and lifetime measurements to address my first and second questions. I have one remaining minor comment related to my first comment. However, my third comment, as detailed below has not yet been adequately addressed and still needs to be resolved before I recommend the publication of current manuscript in Nature Communications.

We thank Reviewer 3 for reviewing our manuscript and for useful comments. Our response to *additional comments and questions* of Reviewer 3 is presented below.

My additional comments and questions:

1. For my first comment, authors performed the temperature-dependent PL measurements in the temperature range of 1.7 K to 10 K. They should also discuss their results in terms of the linewidths of the narrow emitter lines at each position to see whether they are correlated. (Minor)

In SI section Lineshape, we've noted a nearly identical narrow line energy at various x positions. In response to this comment of Reviewer 3, we add a note on a nearly identical narrow line linewidth at various x positions: "The detailed lineshapes show nearly identical energy and linewidth of the narrow line at various x positions."

2. Complementary to the fourth comment of Reviewer 1, to answer also my third comment, authors need to fabricate another sample with (thin ~2nm) or without top hBN layer, and perform one of the characterization techniques I suggested to show the uniformity of the moiré pattern. In that way, they can also show the reproducibility of their results in another sample.

It is not possible for us to make an additional sample with (thin ~2nm) or without top hBN layer, and ... show the uniformity of the moiré pattern. We would like to note that another sample with (thin ~2nm) or without top hBN layer would not have the same low disorder characteristics, in particular, because the hBN layer in our sample protects the MoSe₂/WSe₂ layers of the heterostructure from disorder, thus

allowing *the uniformity*, and removing or substantially thinning the hBN layer would not allow keeping similar low disorder required for both the long-range extension of the narrow PL lines and the efficient IX transport in transport studies in Refs. [36,57,58] in the heterostructure as outlined in our paper. The current characterisation we've already provided within the paper supports our claims: we measured the spatial extension of narrow lines in PL spectra that shows the long-range spatial extension.

It might also help to measure the lifetime of individual narrow PL lines at different positions as suggested before.

The signal is insufficient for these measurements. We noted this in SI: “In these measurements, the signal is insufficient to measure the position-dependent lifetime for a single selected narrow line.”

Reviewer #4 (Remarks to the Author):

The authors have answered all my questions. I am satisfied with the revised manuscript and supplemental material. I would support the publication of the revised version in Nature Communications.

We thank Reviewer 4 for reviewing our manuscript and for useful comments.